# Phylogenetic, Structural and Functional Evolution of the LHC Gene Family in Plant Species

**DOI:** 10.3390/ijms24010488

**Published:** 2022-12-28

**Authors:** Yanhong Lan, Yao Song, Fei Zhao, Yu Cao, Dening Luo, Dairong Qiao, Yi Cao, Hui Xu

**Affiliations:** 1Microbiology and Metabolic Engineering Key Laboratory of Sichuan Province, Key Laboratory of Bio-Resources and Eco-Environment of Ministry of Education, College of Life Sciences, Sichuan University, Chengdu 610065, China; 2School of Automation, Chengdu University of Information Technology, Chengdu 610255, China

**Keywords:** divergence, evolution, expansion, phylogenetic analysis, collinearity analysis, whole-genome duplication

## Abstract

Light-harvesting chlorophyll a/b-binding (LHC) superfamily proteins play a vital role in photosynthesis. Although the physiological and biochemical functions of LHC genes have been well-characterized, the structural evolution and functional differentiation of the products need to be further studied. In this paper, we report the genome-wide identification and phylogenetic analysis of LHC genes in photosynthetic organisms. A total of 1222 non-redundant members of the LHC family were identified from 42 species. According to the phylogenetic clustering of their homologues with *Arabidopsis thaliana*, they can be divided into four subfamilies. In the subsequent evolution of land plants, a whole-genome replication (WGD) event was the driving force for the evolution and expansion of the LHC superfamily, with its copy numbers rapidly increasing in angiosperms. The selection pressure of photosystem II sub-unit S (PsbS) and ferrochelatase (FCII) families were higher than other subfamilies. In addition, the transcriptional expression profiles of LHC gene family members in different tissues and their expression patterns under exogenous abiotic stress conditions significantly differed, and the LHC genes are highly expressed in mature leaves, which is consistent with the conclusion that LHC is mainly involved in the capture and transmission of light energy in photosynthesis. According to the expression pattern and copy number of LHC genes in land plants, we propose different evolutionary trajectories in this gene family. This study provides a basis for understanding the molecular evolutionary characteristics and evolution patterns of plant LHCs.

## 1. Introduction

Light-harvesting chlorophyll a/b binding (LHC) superfamily proteins, as antenna proteins, play an essential role in capturing solar energy, as well as critical roles in plant growth, development, and abiotic stress responses [1]. The LHC superfamily, containing the conserved chlorophyll-binding (CB) domain, is divided into four subfamilies in photosynthetic plants: Lhc, Lil (light-harvesting-like), PsbS, and FCII [2,3]. The Lhc subfamily comprises two groups, named Lhca and Lhcb [4,5]. Similarly, the Lil subfamily is further divided into four subfamilies: SEP (stress-enhanced protein), OHP (one-helix protein), ELIP (early light-induced protein), and Psb33 (photosystem II protein 33) [6,7]. However, The PsbS and FCII subfamilies each include only a single group [8].

Most research on plant LHCs has focused on the Lhc subfamilies Lhca and Lhcb, which play important functions in PSI (photosystem I) and PSII (photosystem II) [9], respectively. The combination of LHC protein with antenna pigments chlorophyll and carotenoids enhances the light capturing ability of plants [10,11]. In Arabidopsis, several Lhc members have been studied in terms of capturing solar energy, seed germination, growth, and plant adaptation to environmental changes [12,13,14]. The overexpression of the tea *CsLhc* gene in the Arabidopsis Lhcb mutant facilitated chlorophyll accumulation and promoted leaf regreening by increasing expression levels of chlorophyll biosynthesis-related genes [15]. The knockout mutant plants of *AtLhcb6* and *AtLhcb5*, as well as *AtLhcb4,* exhibited significantly lower chlorophyll contents in Arabidopsis [16]. When *AcLhcb3.1/3.2* of kiwifruit were over-expressed in tobacco leaves, the results indicated that AcLhcb3.1/3.2 had a higher chlorophyll a content and total chlorophyll content than *AcLhcb1.5* [17]. When *AtLhca2* and *AtLhca3* were over-expressed in bacteria, the results indicated that *AtLhca2* had a higher chlorophyll b content than *AtLhca3* [18]. The heterologous over-expression of tomato *LeLhcb2* in tobacco significantly enhanced the ability of transgenic plants to cope with chilling stress tolerance and alleviated photo-oxidation of PSII [19]. Over-expression of kiwifruit’s *AcLhcb3.1/3.2* in tobacco leaves significantly increased the content of chlorophyll a [17]. In *Gossypium hirsutum*, down-regulated expression of *GhLhcb2.3* by VIGS (virus-induced gene silencing) significantly decreased the content of chlorophyll a. Additionally, *GhLhcb2.3* and *GhLhcb1.4* exhibited complementary functions during total chlorophyll synthesis in photosynthetic light-harvesting [20]. Although a functional characterization of the LHC family has been carried out, it lacks a more comprehensive phylogenetic, structural and functional evolution of the LHC gene family in plant species.

To date, the LHC superfamily has mainly been studied from the perspective of a single species and family [20,21]. However, the evolutionary origin and expansion of the LHC superfamily across the major lineages of green plants have not been reported systematically. With the availability of increasing numbers of genomes, we currently have the ability to study the LHC gene family from a wider perspective. In this study, BLASTP and the domain architecture of Arabidopsis homologues were considered for plant LHC gene superfamily identification at the whole-genome level. The evolutionary relationships, expansion, and expression patterns of different members of the LHC superfamily were investigated. Furthermore, we performed a functional diversity analysis and positive selection analysis, in order to explore the evolution of their structure and function. Based on the research results, we provide a theoretical basis for further research on the function of LHC genes in plants.

## 2. Results

### 2.1. Identification and Distribution of LHC Genes in Plants

To identify putative LHC genes throughout photosynthetic organisms, sequences from 12 Arabidopsis LHC proteins were used as queries in a BLASTp [22] search against 42 species with relatively complete genome annotations, and, as a result, 1222 non-redundant LHC family members and the chromosomal distribution of the LHC genes were identified (Table 1 and Appendix A). The results indicated that LHC superfamily members were distributed in each selected species, and their numbers varied from species to species. The Lhc and Lil subfamilies had the most members, while FCII had the least (Table 1). The FCII subfamily was found in all species, and there were single copies in algae and bacteria, while the members of the Psbs subfamily were only found in green algae and land plants, the Psb33 subfamily genes were only found in land plants, and the tetraploid *Glycine max* genome contained the largest number of LHC genes; meanwhile, the lower alga, plants, and bacteria only contained one member. In the monocots, *Triticum aestivum* had the second-highest number of LHC genes. We observed that the expansion of the LHC superfamily was concentrated in the angiosperms. In the land plants’ genomes, the LHC genes were randomly distributed on chromosomes or scaffolds, and gene numbers varied between 1 and 12 for all chromosomes. In the *Sorghum bicolor* genome, chromosome 2 included the maximum gene number (twelve LHC genes) (Appendix A).

### 2.2. Phylogenetic Analysis and Conserved Motifs of the Plant LHC Gene Family Members

To keep the analysis reliable, an independent unrooted phylogenetic tree was constructed for each subfamily by using LHC full-length proteins to explore the phylogenetic relationship and evolutionary pattern of photosynthetic organisms’ LHC genes. Consistent with the classification of AtLHC proteins, all LHC proteins were grouped into four distinct subfamilies, namely Lhc, Lil, PsbS, and FCII. The results revealed that Lhc, Lil, PsbS, and FCII proteins were clustered into 2, 4, 1, and 1 groups, respectively (Figure 1, Figure 2, Figure 3, Figure 4, Figure 5, Figure 6 and Figure 7, Table 1). To infer structural variations and a possible functional divergence, coding sequences of the LHC genes were analyzed by MEME. The results showed that 24 conserved motifs were obtained, where the type and quantity of motifs were highly variable among subfamilies or groups, and the FCII subfamily contained the highest number of motifs compared to PsbS, Lhc, and Lil (Appendix A). The protein motifs in each sub-group were highly conserved, and some differences were found between different sub-groups. Except for the Psb33 subfamily, motif 3 was widely distributed in all the LHC proteins and located in the conserved CB domain (Figure 1, Figure 2, Figure 3, Figure 4, Figure 5, Figure 6, Figure 7 and Appendix A). Some specific motifs only existed in specific subfamilies. For example, motif 15 was only present in the ELIP subfamily, motif 14 was only present in the PsbS subfamily, and the OHP subfamily only contained motif 18. The LHC subfamilies contained specific motifs, suggesting that they also had specific functions.

### 2.3. Expansion Pattern of LHC Genes and Collinearity Analysis

To further study the duplication events in the evolutionary history of the LHC gene superfamily, *G. max* was used as a model, as it had the largest number of members from this superfamily. According to the species’ evolutionary relationships, we analyzed the genome collinearity among *G. max*, *Populus trichocarpa*, *Vitis vinifera*, and *Amborella trichopoda*, and found that the number of LHC genes increased following the WGDs in these species, including the four subfamilies in the LHC superfamily (Figure 8A).

A total of 31/20 LHC genes in *G. max*/*P. trichocarpa*, 25/17 SPX genes in *P. trichocarpa*/*V. vinifera*, and 14/12 LHC genes in *V. vinifera*/*A. trichopoda* had a high degree of collinearity. The collinear regions for the PsbS and FCII subfamilies were only present in *G. max*/*P. trichocarpa*, the collinear regions for Lil and Lhc subfamilies were present in *G. max*/*P. trichocarpa* and *P. trichocarpa*/*V. vinifera*, and *V. vinifera*/*A. trichopoda* only presented an Lhc collinearity relationship (Figure 8A). However, there were 30 replication events in the *G. max*/ *G. max* genome (Figure 8B).

To further study the conservation of the structure and function of the soybean LHC gene superfamily, the selection pressure was systematically analyzed. The results indicated that there were great differences in selection pressures among different sub-families of the soybean LHCs’ superfamily. In addition, the Ks value of duplicated gene pairs varied from 0.02 to 1.67 (Figure 8C), suggesting that duplicated pairs have different evolution rates. The higher Ks value of dispersed duplication when compared to whole-genome duplication (WGD) suggested that it may have happened earlier than the WGD event. Most duplicated gene pairs underwent WGD, indicating that WGD played a vital role in expanding LHCs in soybeans. The Ka/Ks of the PsbS gene family was the lowest (the average was 0.175, less than 1), followed by FCII (Ka/Ks was 0.384), suggesting that the selection pressure on these two subfamilies was high. The Ka/Ks ratios for LHC and Lil were higher (an average of 1.077 and 1.067, respectively), suggesting that the genes in these two gene families may have experienced positive selection (Figure 8C).

### 2.4. Expression Patterns of LHCs in Plants

To further study the potential functions of the LHC superfamily, the issue-specific and abiotic stress expression patterns of several representative LHC genes from *Oryza sativa*, model *A. thaliana*, and *G. max* were analyzed, using microarray data of LHC genes from the BAR database across different tissue types (e.g., root, leaf, flower, and SAM) and in response to abiotic stresses (e.g., drought, salt, light, heat, and cold).

The results indicated that 18 LHC genes were expressed in leaf tissues, with a strong tissue-specific expression pattern. Lil genes were highly expressed in leaves, such as Glyma.16G163800.1.p, Glyma.04G249700.1.p, Glyma.03G262300.2.p, LOC_Os01g41710, and LOC_Os01g64960, suggesting that these genes have important functions in leaves. For the Lil gene in soybeans (Glyma.03G262300.2.p), the highest expression was detected in flowers, and the lowest expression was in the root, suggesting that this gene may play a key role in flower development (Figure 9A–C). *A. thaliana* Lil genes (AT4G14690 and AT3G22840) were highly expressed in shoot and root tissues under cold stress, and their expression levels were relatively high under salt and UV-B stress in shoot tissues (Figure 9D–E); however, the expression of these two genes was low under drought and heat stress. For LHC genes in rice, the expression was low under abiotic stress, among which LOC_Os07g08150, LOC_Os01g14410, and LOC_Os04g59440 had the lowest expression under cold stress, suggesting that these genes were functional in the plant response to these stimuli. LOC_Os01g41710 was expressed highly in tissues, with no significant change under abiotic stress.

## 3. Discussion

Light-harvesting chlorophyll a/b binding protein (LHC) is the most abundant protein complex on the thylakoid membrane, and it plays an important role in plant growth and development (e.g., capturing and transforming light during photosynthesis and having an association with oxidative stress) [23,24,25]. Members of the LHC superfamily have been identified and characterized in land plants, but there has still no systematic and comprehensive report on the photosynthetic LHC superfamily organism. In the present study, we performed a genome-wide analysis of LHC gene family members in 42 species. Phylogenetic analysis indicated that the LHC superfamily was mainly divided into four subfamilies (Lhc, Lil, PsbS, and FCII), where the genetic origins of these subfamilies were inconsistent and each type had experienced a long-term independent evolution process. The CB domain was highly conserved in the LHC family in different plants [26]. The topology of the evolutionary tree indicated that LHC genes from all land plants are generally conserved and that they share a similar pattern in terms of protein domain and motifs. Bacteria and algal LHC were distantly associated with those of land plants, as supported by differences in the number and organization of conserved protein motifs.

Researchers have reported that the duplication, loss, pseudogene, and other changes of gene family members during the process of evolution are common phenomena [27]. Genomic replication often results in an expansion of the gene family [28,29,30]. In this study, we performed a synteny analysis of Lhc superfamily genes in *G. max*, *P. trichocarpa*, *V. vinifera*, and *A. trichopoda* (Figure 8). Genome-wide replication analysis showed that LHC gene family members experienced multiple WGD events between genomes. However, the difference in the number of LHC members in the superfamily and subfamilies of the plants suggested that they may have experienced different evolution patterns. In addition, an analysis of the selection pressure in *G. max* indicated that the LHC gene family had different evolutionary directions due to different selection pressures during its evolution, thus forming a diverse population. The average gene Ka/Ks ratio for the Lhc and FCII subfamilies was more than 1, suggesting that the gene family was greatly affected by positive selection, while those of the other gene families were all under 1, suggesting that the gene family has been greatly affected by negative selection [17]. After gene fragment replication, a strong purification selection was experienced. These results indicate that the genes of the PsbS and Lil subfamilies are relatively conservative in structure.

The expression analysis demonstrated that multiple LHC genes were specifically expressed in different tissues and stresses; however, the differentiation of expression patterns was mainly concentrated among different subfamily genes. With an increase in the number of gene family members in the species, the differentiation of expression patterns was more obvious [31]. The expression of soybean LHC family genes was the highest in mature leaves, followed by flowers, and it was the lowest in roots; this distribution was consistent with the conclusion that LHC—as a light-harvesting pigment-binding protein gene—is expressed in the thylakoids of chloroplasts, the light reaction site of photosynthesis. Photosynthesis mainly occurs in the leaves of plants [32]. A large amount of light energy is absorbed by the leaves, while the root hardly receives light. LOC_ Os01g41710 is highly expressed in the mature leaves of rice, but it is hardly expressed in roots, suggesting that this gene may be necessary for plant photosynthesis. Analysis of the response of LHC family members to abiotic stress indicated that the genes of LHC family members in Arabidopsis were expressed at low levels under stress; therefore, stress treatment would inhibit plant capture light efficiency and affect photosynthesis. Meanwhile, rice LHC family members were highly expressed under cold stress, suggesting that this gene has a functional role in the plant response to such stimuli.

Overall, 1222 LHC genes from 42 species were comprehensively and systematically for the first time, to the best of our knowledge. The LHC superfamily is divided into four subfamilies, Lhc, Lil, PsbS, and FCII, according to the phylogeny of its homologues with *A. thaliana*. The results of the whole-genome replication event indicated that the molecular evolution of LHC has been characterized by large-scale indirect amplification and gene loss. The number of LHC gene members was the largest in soybeans, and the molecular evolution of the LHC family in the genome showed indirect amplification and gene loss. LHC genes were purified and screened, but a strong selectivity was observed in the Lhc gene subfamily. The tissue and stress-condition-specific expression patterns indicated that LHCs play important roles in multiple biological processes. This study provided novel viewpoints regarding the functional evolution and expansion of the LHC gene family in various important species.

## 4. Methods and Materials

### 4.1. Identification of LHC Gene Family Members

In order to explore the evolutionary divergence and expansion of the LHC superfamily across photosynthetic organisms, 42 representative species with relatively complete annotated genome data were used as research subjects for the assessment of the LHCs’ taxonomy and phylogenetic relationships. The genomic data were obtained from the Plant JGI Database phytozome v12.1 (https://phytozome.jgi.doe.gov/pz/portal.html (accessed on 6 October 2022)) [33] and National Center for Biotechnology Information Genome database (https://ncbi.nlm.nih.gov/genome/ (accessed on 22 December 2022)). Then, twelve previously identified and characterized LHC proteins from *Arabidopsis thaliana* [21] (GenBank: Lhca1, AT3G54890; Lhca2, AT3G61470; Lhcb1, AT1G29920; Lhcb3, AT5G54270; ELIP1, AT3G22840; ELIP2, AT4G14690; SEP1, AT4G34190; SEP2, AT2G21970; OHP1, AT5G02120; OHP2, AT1G34000; Psb33, AT1G71500; PsbS, AT1G44575) were employed to construct a query protein set. The candidate LHCs were identified using BLASTp (BLAST plus: architecture and applications) [22], with a cutoff score of ≥100 and an e-value of ≤ 1 × e ^−10^. The Conserved Domain Database (CDD) (https://ncbi.nlm.nih.gov/cdd (accessed on 8 January 2020)) [34], Pfam [35], and SMART (http://smart.embl-heidelberg.de (accessed on 26 October 2020)) [36] were used to confirm the identity of LHC proteins containing a chlorophyll-binding (CB) domain. The location of all species’ LHC genes was extracted from the corresponding GFF file using an in-house Perl script.

### 4.2. Phylogeny Analysis and Identification of Conserved Motifs

Multiple sequence alignments of the amino acid sequences of identified LHC proteins were performed using ClustalW and ClustalX with default parameters [37] Unrooted phylogenetic trees were constructed based on the neighbor-joining (NJ) method with a JTT (Jones–Taylor–Thornton) model and 1000 bootstrap replicates, using the MEGA software (v7.0.26, download from https://www.megasoftware.net/ (accessed on 9 August 2021)) [38]. The phylogenetic trees were visualized using FigTree v1.4.3 (http://tree.bio.ed.ac.uk/ (accessed on 13 June 2019)). To obtain insight into the divergence and function of the LHC proteins, the twelve most conserved motifs were identified using the MEME tool (http://meme-suite.org/tools/meme (accessed on 17 December 2022)) [39], and the results were imported to the TBtools [40] to visualize protein structures.

### 4.3. Microarray Based Expression Analysis

The original RNA-Seq expression data of LHC family members from different tissues and abiotic stress were obtained from BARPB (Bio-Analytic Resource for Plant Biology http://bar.utoronto.ca/ (accessed on 7 January 2022)). Subsequently, the BAR HeatMapper Plus tool (http://bar.utoronto.ca/ntools/cgi-bin/ntools_heatmapper_plus.cgi (accessed on 7 January 2022)) was used to analyze the expression levels of selected plant LHC genes.

### 4.4. Gene Duplication and Synteny Analysis

The MCScanX [41] software was used to investigate the synteny, collinearity, and gene duplication patterns of plant LHCs with default parameters. TBtools [40] was used to calculate the synonymous (Ks) and nonsynonymous (Ka) mutation rates of the duplicated LHC gene pairs [42], and the Ka/Ks ratio was calculated to measure the selection pressure (Ka/Ks < 1, purifying; or Ka/Ks > 1, positive) during evolution [43]. The syntenic relationships between identified genes were visualized using TBtools.

## Figures and Tables

**Figure 1 ijms-24-00488-f001:**
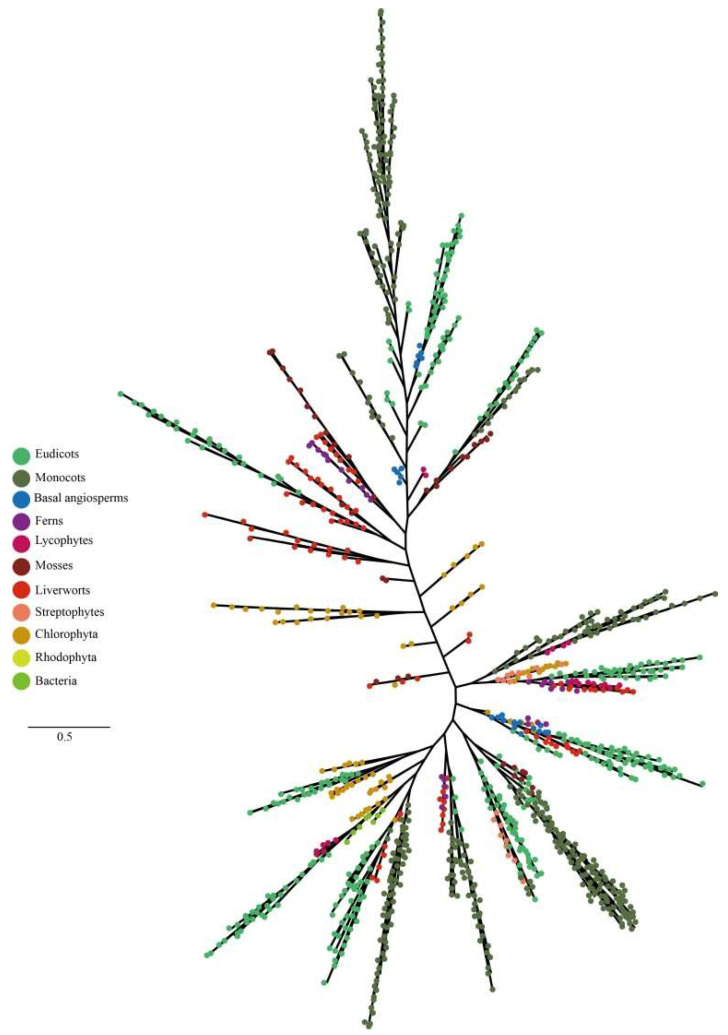
Phylogenetic analysis of Lhc gene subfamily from 42 species.

**Figure 2 ijms-24-00488-f002:**
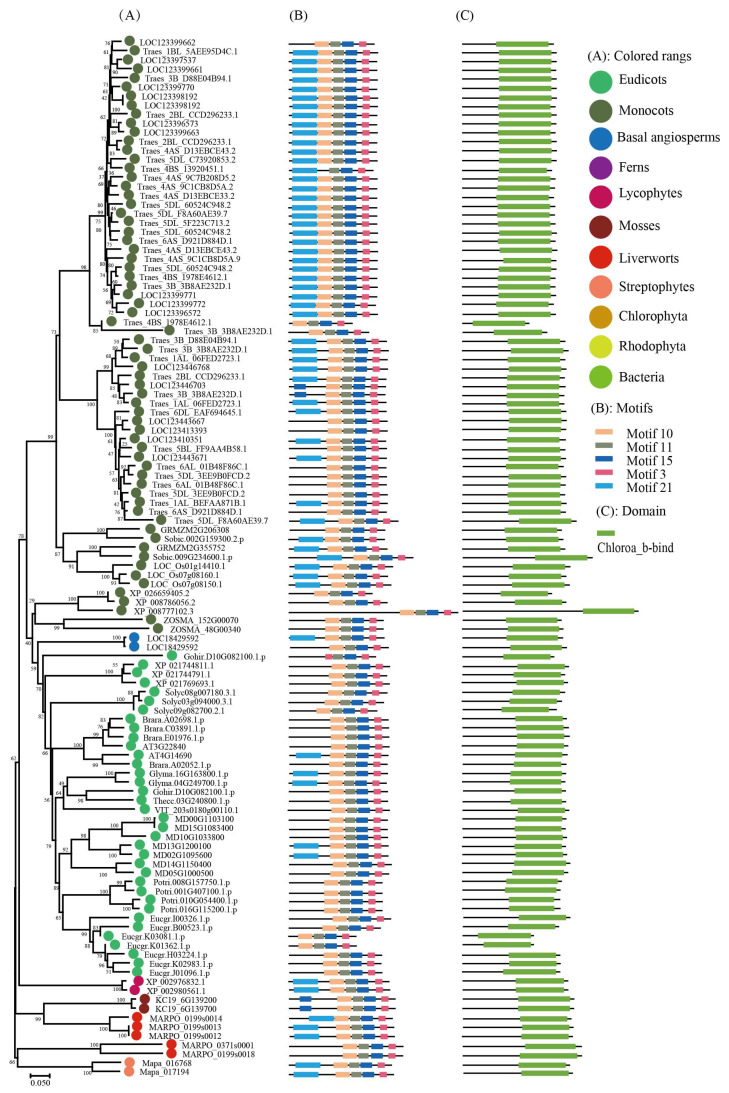
Phylogenetic evolutionary tree, protein motifs, and domains of ELIP subfamily members: (**A**) Unrooted phylogenetic tree of the ELIP subfamily; (**B**) motifs of the ELIP proteins. The different colors indicate five motifs (left panel); (**C**) protein domains of the ELIP genes.

**Figure 3 ijms-24-00488-f003:**
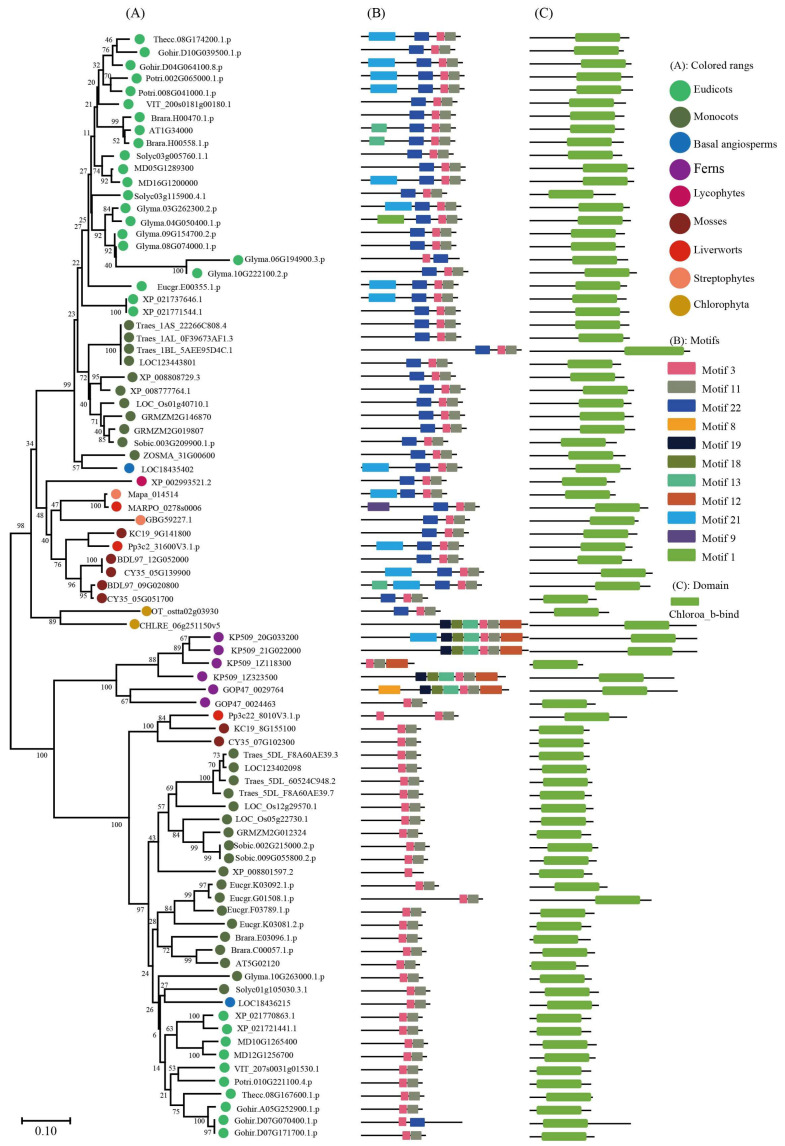
Phylogenetic evolutionary tree, protein motifs, and domains of the OHP subfamily members: (**A**) Unrooted phylogenetic tree of the OHP subfamily; (**B**) motifs of the OHP proteins. The different colors indicate 11 motifs (left panel); and (**C**) protein domains of the OHP genes.

**Figure 4 ijms-24-00488-f004:**
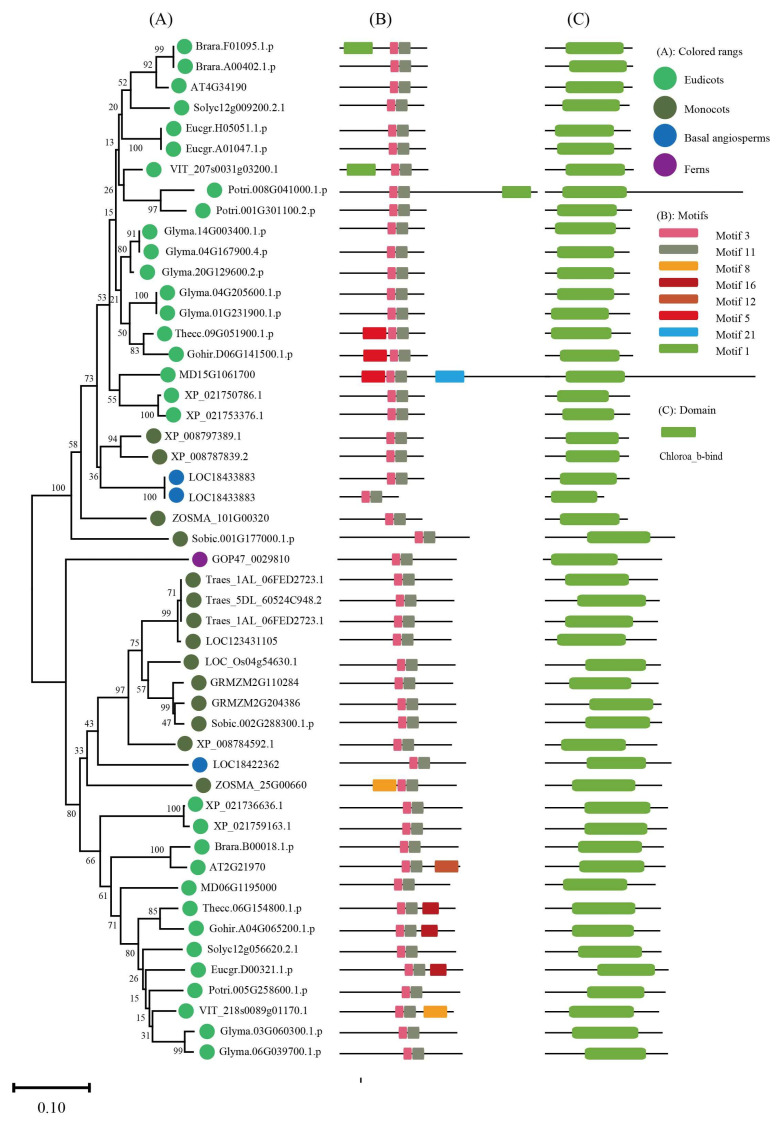
Phylogenetic evolutionary tree, protein motifs, and domains of the SEP subfamily members: (**A**) Unrooted phylogenetic tree of the SEP subfamily; (**B**) motifs of the SEP proteins. The different colors indicate eight motifs (left panel); (**C**) protein domains of the SEP genes.

**Figure 5 ijms-24-00488-f005:**
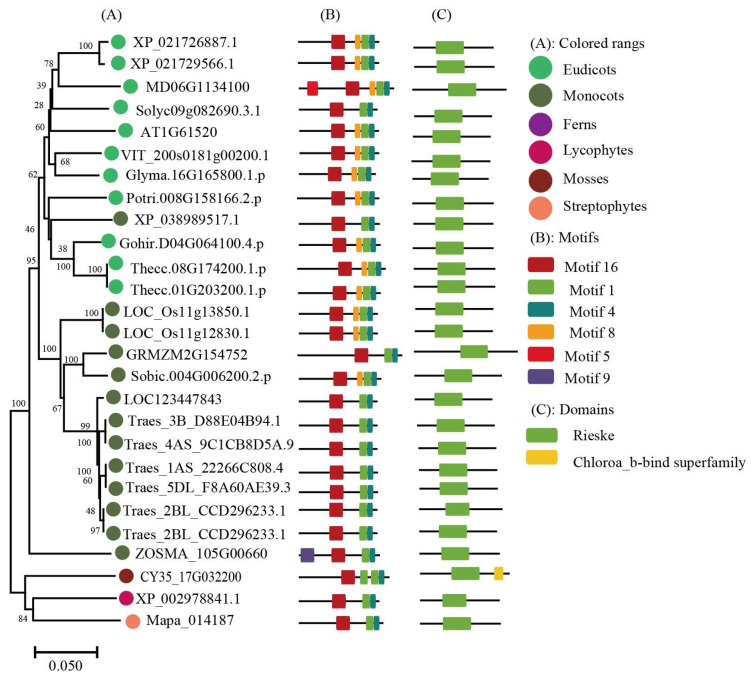
Phylogenetic evolutionary tree, protein motifs, and domains of the Psb33 subfamily members: (**A**) Unrooted phylogenetic tree of the Psb33 subfamily; (**B**) motifs of the Psb33 proteins. The different colors indicate six motifs (left panel); (**C**) protein domains of the Psb33 genes.

**Figure 6 ijms-24-00488-f006:**
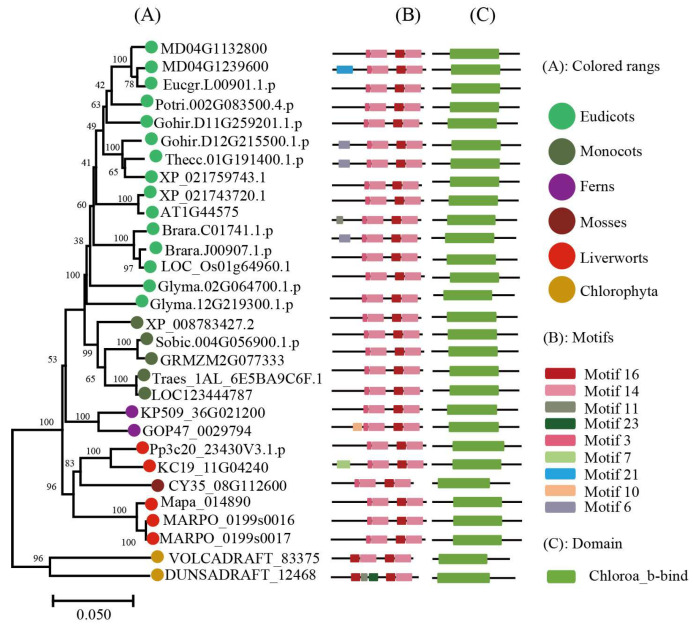
Phylogenetic evolutionary tree, protein motifs, and domains of the PsbS subfamily members: (**A**) Unrooted phylogenetic tree of the PsbS subfamily; (**B**) motifs of the PsbS proteins. The different colors indicate nine motifs (left panel); (**C**) protein domains of the PsbS genes.

**Figure 7 ijms-24-00488-f007:**
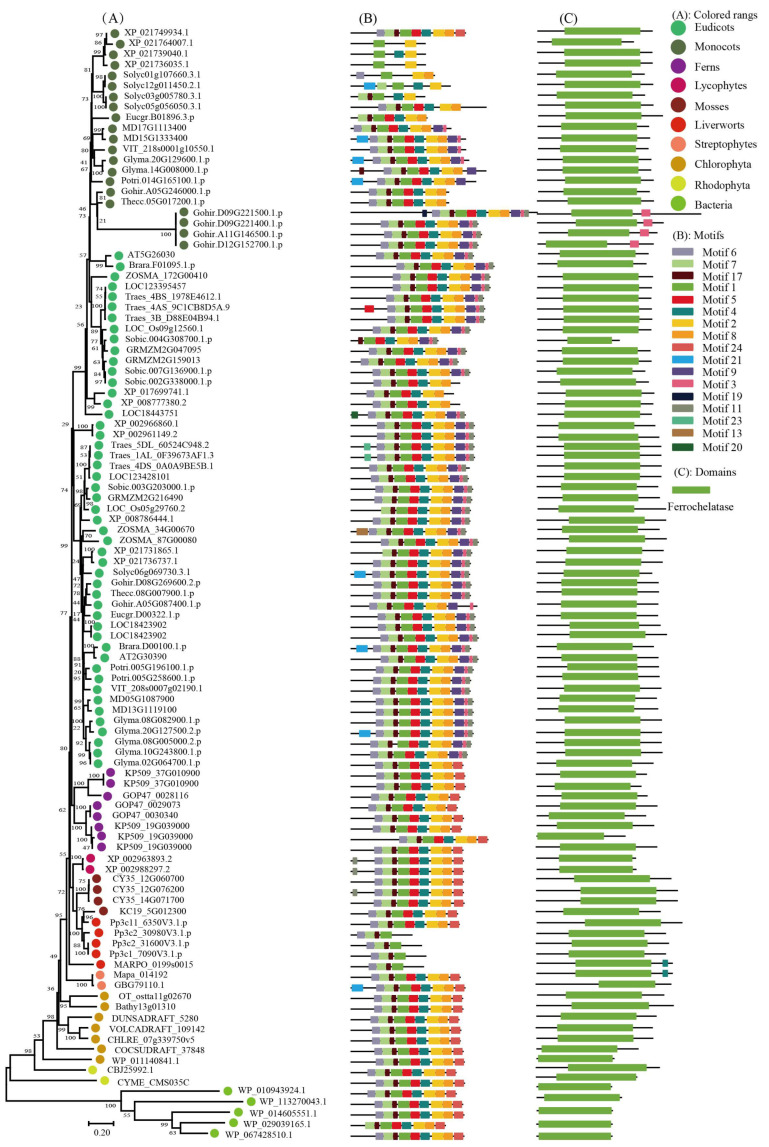
Phylogenetic evolutionary tree, protein motifs, and domains of the FCII subfamily members: (**A**) Unrooted phylogenetic tree of the FCII subfamily; (**B**) motifs of the FCII proteins. The different colors indicate seventeen motifs (left panel); (**C**) protein domains of the FCII genes.

**Figure 8 ijms-24-00488-f008:**
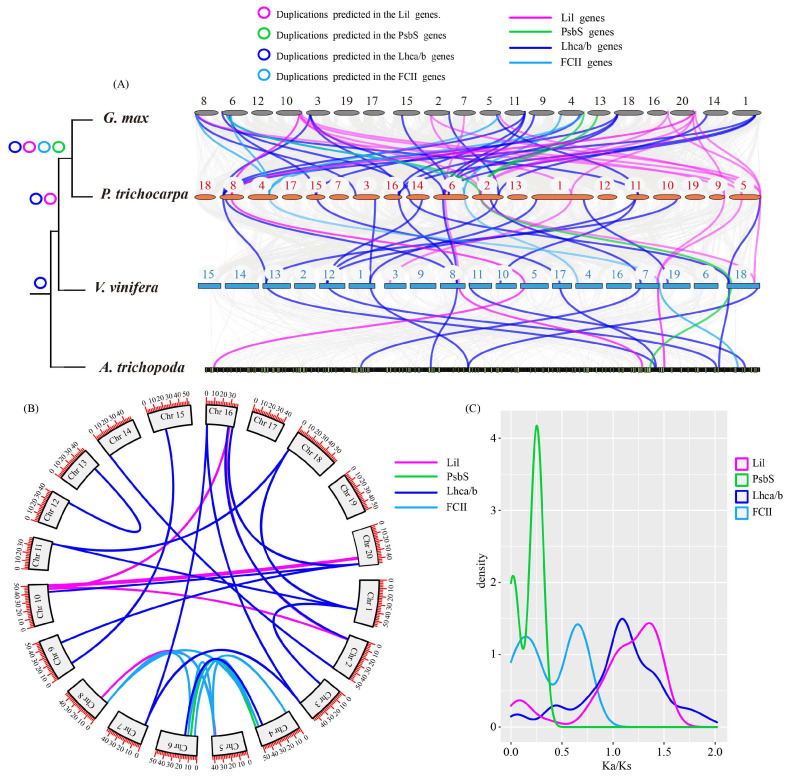
Duplication events of LHC genes in *G. max*. (**A**) Syntenic relationships of LHC genes among *Glycine max*, *Populus trichocarpa*, *Vitis vinifera*, and *Amborella trichopoda*; (**B**) syntenic relationships of LHC genes in *Glycine max/ Glycine max*; (**C**) distribution of Ka/Ks values among LHC genes in *Glycine max*.

**Figure 9 ijms-24-00488-f009:**
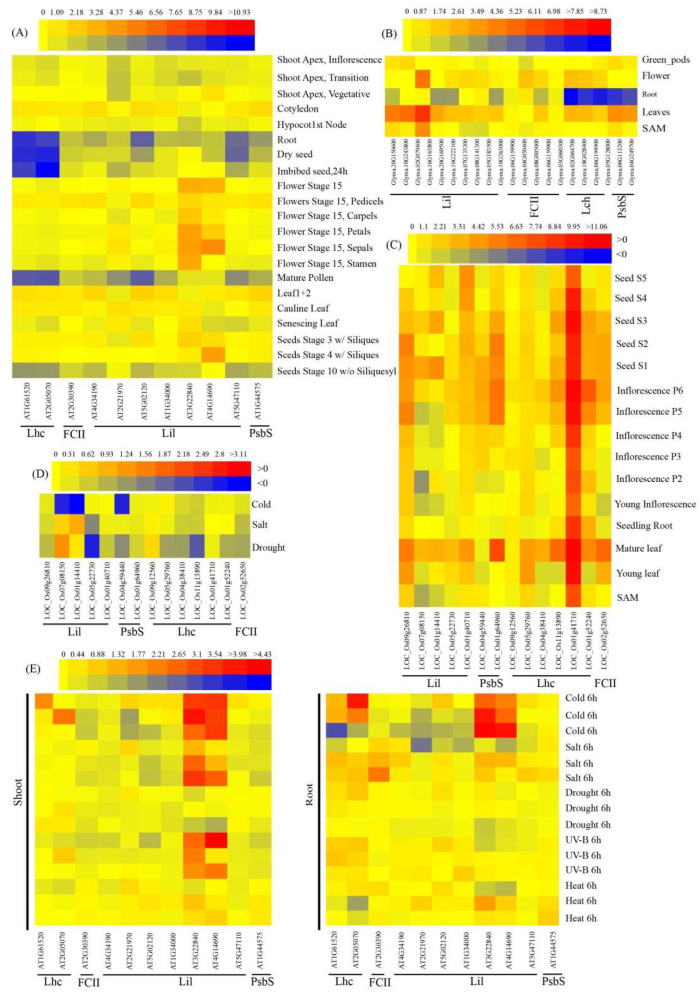
Expression profiles of LHC genes in representative species: (**A**–**C**) Spatial and temporal expressions of LHC genes from *Arabidopsis*, *Glycine max* (soybean), and *Oryza sativa* (rice); (**D**,**E**) Expression of LHC genes under different stress treatments.

**Table 1 ijms-24-00488-t001:** Distribution of the members of the LHC gene family in 42 plant species.

Taxa	Species	Total	Lhca/b	Lil				PsbS	FCII
				OHP	SEP	ELIP	Psb33		
Eudictos	Arabidopsis thaliana	32	22	2	2	2	1	1	2
	Brassica rapa	44	29	4	3	4	0	2	2
	Gossypium raimondii	44	24	5	2	3	1	2	7
	Theobroma cacao	25	15	2	2	1	2	1	2
	Eucalyptus grandis	30	17	5	3	2	0	1	2
	Malus domestica	42	26	4	2	3	1	2	4
	Glycine max	62	34	7	7	3	1	3	7
	Populus trichocarpa	36	22	3	3	3	1	1	3
	Vitis vinifera	31	20	2	2	3	1	1	2
	Solanum lycopersicum	40	24	3	2	4	1	1	5
	Chenopodium quinoa	49	29	4	4	2	2	2	6
Monocots	Zostera marina	32	22	1	2	2	1	1	3
	Phoenix dactylifera	33	21	3	3	2	0	1	3
	Triticum aestivum	58	31	6	3	3	6	3	6
	Hordeum vulgare	30	22	2	1	1	1	1	2
	Zea mays	37	25	3	2	2	1	1	3
	Sorghum bicolor	29	17	3	2	1	1	1	4
	Oryza sativa	25	14	3	1	1	2	2	2
Basal angiosperms	Amborella trichopoda	27	15	2	3	3	0	1	3
Ferns	Adiantum capillusveneris	42	35	2	0	1	0	1	3
	Ceratopteris richardii	54	40	4	1	2	0	2	5
Lycophytes	Selaginella moellendorffii	25	16	1	0	2	0	2	4
Mosses	Sphagnum magellanicum	53	37	5	0	4	0	4	3
	Ceratodon purpureus	41	34	2	0	3	0	1	1
Liverworts	Marchantia polymorpha	54	45	2	0	2	0	3	2
	Physcomitrium patens	51	40	2	0	4	0	1	4
Streptophyta	Marchantia paleacea	56	47	2	0	2	1	3	1
	Chara braunii	16	12	1	0	0	0	2	1
Chlorophyta	Chlamydomonas reinhardtii	28	23	1	0	1	0	2	1
	Volvox cartei	30	27	0	1	0	0	1	1
	Ostreococcus sp	11	9	0	0	1	0	0	1
	Coccomyxa subellipsoidea	22	20	0	0	1	0	0	1
	Dunaliella salina	16	13	0	0	1	0	1	1
Rhodophyta	Cyanidioschyzon merolae	1	0	0	0	0	0	0	1
	Ectocarpus siliculosus	9	8	0	0	0	0	0	1
Bacteria	Bathycoccus prasinos	1	0	0	0	0	0	0	1
	Panttoea ananatis	1	0	0	0	0	0	0	1
	Erwinia gerundensis	1	0	0	0	0	0	0	1
	Halomonas sulfidaeris	1	0	0	0	0	0	0	1
	Cronobacter sakazakii	1	0	0	0	0	0	0	1
	Geobacter sulfurreducens	1	0	0	0	0	0	0	1
	Gloeobacter violaceus	1	0	0	0	0	0	0	1

## Data Availability

Not applicable.

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
