# Peer review of "Phylogenetic, Structural and Functional Evolution of the LHC Gene Family in Plant Species"

_ijms, 2022, doi:10.3390/ijms24010488_

Round 1
Reviewer 1 Report (Previous Reviewer 3)
The main lack of the ms is that is a descriptive one. Even in the title it says nothing about what was discovered here and what new this ms is telling us.
The introduction is still specifically telling about several mutants. Nothing about the functioning of LHC proteins as the inner and outer antenna, the binding of chlorophyll and carotenoids and the family of stress related LHC like proteins.
Author Response
- Reviewer 1:
The introduction is still specifically telling about several mutants. Nothing about the functioning of LHC proteins as the inner and outer antenna, the binding of chlorophyll and carotenoids and the family of stress related LHC like proteins.
Corrected.
The combination of LHC protein with antenna pigments chlorophyll and carotenoids enhances the light capture ability of plants [10]. In Arabidopsis, several Lhc members have been studied, in terms of capturing solar energy, seed germination, and growth, and plant adaptation to environmental changes [12-14]. The overexpression of the tea CsLhc gene in Arabidopsis Lhcb mutant facilitated chlorophyll accumulation and promoted leaf regreen by increasing expression levels of chlorophyll biosynthesis-relative genes [15]. The knockout mutant plants of AtLhcb6 and AtLhcb5, and AtLhcb4 exhibited significantly lower chlorophyll contents in Arabidopsis [16]. When AcLhcb3.1/3.2 of the Kiwifruit were over-expressed in tobacco leaves, the results indicated that AcLhcb3.1/3.2 had higher chlorophyll a content, and total chlorophyll content than AcLhcb1.5 [17]. When AtLhca2 and AtLhca3 were over-expressed in bacteria, the results indicated that AtLhca2 had higher chlorophyll b content than AtLhca3 [18]. Heterologous over-expression of tomato LeLhcb2 in tobacco significantly enhanced the ability of transgenic plants to cope with chilling stress tolerance and alleviated photo-oxidation of PSII [19]. Over-expression of Kiwifruit AcLhcb3.1/3.2 in tobacco leaves significantly increased the content of chlorophyll a [20].
Reviewer 2 Report (Previous Reviewer 1)
I still suggest the authors should replaced the genome information of 42 species with the new version of genome. The gene number should not be written as XP... or NP.....
Author Response
- Reviewer 2 :
“I still suggest the authors should replaced the genome information of 42 species with the new version of genome. The gene number should not be written as XP... or NP.....”
Corrected. We replaced the genome information of 42 species with the new version of genome in the supplementary table.
This manuscript is a resubmission of an earlier submission. The following is a list of the peer review reports and author responses from that submission.
Round 1
Reviewer 1 Report
This paper discribed that phylogenetic, structure and functional evolution of the LHC gene family in plant species. I think this paper is of great interest to the plant scientific community. However, there are both insignificant and significant remarks, as well as issues that should be taken into account.
1. Why the authors choose G. max , Populus trichocarpa, Vitis vinifera, Amborella trichopoda for the collinearity analysis of LHC family genes.
2. As I know, a lot of plant species genomes have been updated with new developed sequencing technology. Whether the new versions of genome information of 42 plant species were used for the identification of LHC family genes? Why most of the LHC genes were written as XP... NP...?
3. Arabidopsis thaliana should be written as italic, revise it throught the paper.
4. Line 31 [1. is wrong? should be removed.
5. Lines 32-39, the citations of references is not shown.
Author Response
- Reviewer 1 :
- Why the authors choose G. max, Populus trichocarpa, Vitis vinifera, Amborella trichopoda for the collinearity analysis of LHC family genes.
Corrected. G. max was used as a model, as it had the largest number of members from LHC superfamily. According to species evolutionary relationships, we analyzed the genome collinearity among G. max, Populus trichocarpa, Vitis vinifera, and Amborella trichopoda.
- As I know, a lot of plant species genomes have been updated with new developed sequencing technology. Whether the new versions of genome information of 42 plant species were used for the identification of LHC family genes? Why most of the LHC genes were written as XP... NP...?
Corrected. The new versions of genome information of 42 plant species were used for the identification of LHC family genes. There is a corresponding genome ID number in the NCBI public database, and the sequence protein can be quickly queried.
- Arabidopsis thaliana should be written as italic, revise it throught the paper.
Corrected. “Arabidopsis thaliana” is replaced with “Arabidopsis thaliana” in our manuscript.
- Line 31 [1. is wrong? should be removed.
Corrected. Line 31 “[1” is replaced with “[1]”.
- Lines 32-39, the citations of references is not shown.
Corrected. Lines 32-39, the citations of references is shown.
Reviewer 2 Report
The manuscript "Phylogenetic, structure and functional evolution of the LHC gene family in plant species" reports the genome wide identification and phylogenetic analysis of LHC genes in plants. This study did a lot of data mining work and indicated that the evolution and expansion of 1222 LHC superfamily members from 42 plant species were drove by the whole genome replication (WGD) event. Although the study is a knowledge updating case, it may be of interest to researchers working in this filed. Addressing the following points would somehow improve its quality.
1. A major problem, the writing format of this article needs to be carefully corrected and updated. Some citation failure and citation inappropriate format problems were appeared in many places (Lin31-39,Lin247-248,Lin280-282). Please the corresponding author strictly control the writing quality.
2. Authors indicated this study lay a foundation for its functional characterization, are the authors doing any functional analyses experiments on LHC family?
3. In abstract, Lin15-17, it is suggested that the authors rewrote the relevant statements according to the results and conclusions of the text, e.g. lin233-234.
4. In Table1, it is suggested that the authors add some notes to point out the meaning that the circle size and colors represented
5. In Figure1, it is suggested that the authors update figure and point out different cluster groups, as well as add the corresponding figure note.
6. In Lin131-132, it is suggested that the authors add “Table 1” in this sentence to do the data support.
7. In Figure8, it is suggested that the authors update plate numbers of plate A and split each plate to ensure that it is easy for readers to understand.
8. In Lin 160, Lin273-274, Lin 282, please authors add a citation for the software/online tools.
9. In LIn197-205, Authors indicated that LHC gene family members experienced multiple WGD events and they might have experienced different evolution patterns. it is suggested that the authors add some discussion about the impact of the whole genome duplication event on the amplification of the LHC gene family members.
Finally, authors should double check the writing typos in the article and be consistent with the writing standard and style of the abbreviations, supplement figure and table.
Author Response
- Reviewer 2 :
- A major problem, the writing format of this article needs to be carefully corrected and updated. Some citation failure and citation inappropriate format problems were appeared in many places (Lin31-39, Lin247-248, Lin280-282). Please the corresponding author strictly control the writing quality.
Corrected. The citations of references is shown.
- Authors indicated this study lay a foundation for its functional characterization, are the authors doing any functional analyses experiments on LHC family?
Corrected. “This study provides a basis for understanding the molecular evolutionary characteristics and evolution patterns of plant LHCs.” is added.
- In abstract, Lin15-17, it is suggested that the authors rewrote the relevant statements according to the results and conclusions of the text, e.g. lin233-234.
Corrected. “In addition, the transcriptional expression profiles of LHC gene family members in different tissues and their expression patterns under exogenous abiotic stress conditions significantly differed, and the LHC genes are highly expressed in mature leaves, which are consistent with the conclusion that LHC is mainly involved in the capture and transmission of light energy in photosynthesis.”
- In Table1, it is suggested that the authors add some notes to point out the meaning that the circle size and colors represented
Corrected.
- In Figure1, it is suggested that the authors update figure and point out different cluster groups, as well as add the corresponding figure note.
Corrected.
- In Lin131-132, it is suggested that the authors add “Table 1” in this sentence to do the data support.
Corrected.
- In Figure8, it is suggested that the authors update plate numbers of plate A and split each plate to ensure that it is easy for readers to understand.
Corrected.
- In Lin 160, Lin273-274, Lin 282, please authors add a citation for the software/online tools.
Corrected. The citations of references are added.
- In LIn197-205, Authors indicated that LHC gene family members experienced multiple WGD events and they might have experienced different evolution patterns. it is suggested that the authors add some discussion about the impact of the whole genome duplication event on the amplification of the LHC gene family members.
Corrected.
Reviewer 3 Report
In this manuscript, a bioinformatic analysis of the amino acid sequences of the LHC superfamily has been performed. In total, 1222 amino acid sequences derived from the corresponding genes of 42 species were compared and a brief phylogenetic analysis carried out. The manuscript is a list of all this sequences divided into groups. In additional, a prediction about the origin of the present genes by gene duplication is shown, as well as an analysis of gene expression taken from the data in the public data base of gene expression in Arabidopsis. All in all, this is a bioinformatical analysis of the LHC sequences that is mostly known and published. The manuscript lacks the group division according to the pigments binding of each group which is an important parameter to consider. There is also only a little description of the different functions of the groups and how these effected their structure and amino acid sequence.
The manuscript should be reviewed for a better English and scientific writing. Some examples:
Abstract: The structure and function of the gene products (not gene). “still unclear” not true. PsbS and FCII were not purified in this work. “indicating their have multiple biological significance” this is too far going statement. LHCs (last sentence).
Introduction: The firs references are not listed.
The second paragraph is a mix of the conclusions of several papers analyzing LHCs with any rational or logic why they are listed here. There is no a scientific question developed here.
Also here, the statements “until now there was not ….” “have not been reported ….. “ should be more accurate.
Higher plants is not the useful scientific version. Use land plants.
“indicating that this gene may play a key role in flower development”. “Suggesting that”
There are many places were s is missing from plural. Is instead of are. etc
Author Response
- Reviewer 3 :
- Abstract: The structure and function of the gene products (not gene). “still unclear” not true. PsbS and FCII were not purified in this work. “indicating their have multiple biological significance” this is too far going statement. LHCs (last sentence).
Corrected. “The structure and function of the gene products (not gene); still unclear” is replaced with “Although the physiological and biochemical functions of LHC genes have been well-characterized, the structural evolution and functional differentiation of the products need be further studied.”; “PsbS and FCII were not purified in this work.” Is replaced with “The selection pressure of photosystem II sub-unit S (PsbS) and ferrochelatase (FCII) families were higher than other subfamilies.”; “indicating their have multiple biological significance” is replaced with “In addition, the transcriptional expression profiles of LHC gene family members in different tissues and in response to exogenous abiotic stress conditions indicated functional similarity of members in the same clade.”
- Introduction: The first references are not listed.
Corrected.
3.The second paragraph is a mix of the conclusions of several papers analyzing LHCs with any rational or logic why they are listed here. There is no a scientific question developed here.
Corrected. “Although functional characterization of LHC family has been carried out, it lacks more comprehensive phylogenetic, structural and functional evolution of the LHC gene family in plant species.” is added in the second paragraph.
- Also here, the statements “until now there was not ….” “have not been reported ….. “ should be more accurate.
Corrected. “until now there was not ” is replaced with “have not been reported”
- Higher plants is not the useful scientific version. Use land plants.
Corrected. “Higher plants” is replaced with “land plants”.
- “indicating that this gene may play a key role in flower development”. “Suggesting that”
Corrected.
- There are many places were s is missing from plural. Is instead of are. etc
Corrected.
Round 2
Reviewer 2 Report
Many thanks to authors for responses. The main idea of the manuscript was made clearer and more complete. Although I have cautious about some responses,this study was no less than a timely update. Now,I agree to accept this MS.